# Highly Sensitive and Stable Multifunctional Self-Powered Triboelectric Sensor Utilizing Mo_2_CT_x_/PDMS Composite Film for Pressure Sensing and Non-Contact Sensing

**DOI:** 10.3390/nano14050428

**Published:** 2024-02-27

**Authors:** Jialiang Fan, Chenxing Wang, Bo Wang, Bin Wang, Fangmeng Liu

**Affiliations:** State Key Laboratory of Integrated Optoelectronics, Key Laboratory of Advanced Gas Sensors, Jilin Province, College of Electronic Science and Engineering, Jilin University, 2699 Qianjin Street, Changchun 130012, China; jlfan21@mails.jlu.edu.cn (J.F.);

**Keywords:** non-contact sensor, triboelectric nanogenerator, Mo_2_CT_x_, self-powered

## Abstract

Sensors based on triboelectric nanogenerators (TENGs) are increasingly gaining attention because of their self-powered capabilities and excellent sensing performance. In this work, we report a Mo_2_CT_x_-based triboelectric sensor (Mo-TES) consisting of a Mo_2_CT_x_/polydimethylsiloxane (PDMS) composite film. The impact of the mass fraction (wt%) and force of Mo_2_CT_x_ particles on the output performance of Mo-TES was systematically explored. When Mo_2_CT_x_ particles is 3 wt%, Mo-TES3 achieves an open-circuit voltage of 86.89 V, a short-circuit current of 578.12 nA, and a power density of 12.45 μW/cm^2^. It also demonstrates the ability to charge capacitors with varying capacitance values. Additionally, the Mo-TES3 demonstrates greater sensitivity than the Mo-TES0 and a faster recovery time of 78 ms. Meanwhile, the Mo-TES3 also demonstrates excellent stability in water washing and antifatigue testing. This demonstrates the effectiveness of Mo-TES as a pressure sensor. Furthermore, leveraging the principle of electrostatic induction, the triboelectric sensor has the potential to achieve non-contact sensing, making it a promising candidate for disease prevention and safety protection.

## 1. Introduction

With the rise of technologies such as 5G and IoT, the world has entered the age of smart information [1,2,3,4]. Sensors, as the primary means of gathering information, play an indispensable role across all aspects of life [5,6,7]. To meet the growing demand for technical capabilities, sensors must not only capture information more reliably and accurately but also adapt to diverse application scenarios [8,9,10]. A significant issue with modern smart devices is the need for traditional sensors to be powered. However, traditional power sources, such as batteries and capacitors, have limitations due to their inflexible structure and the need for frequent charging and maintenance. As a result, they are unsuitable for meeting the energy supply requirements of modern smart devices. The COVID-19 pandemic has underscored the urgency of containing the spread of bacteria and viruses, propelling non-contact sensors into the spotlight due to their capacity to transmit signals without physical contact [11]. Traditional non-contact sensors, such as inductive sensors, capacitive sensors, and photoelectric sensors, are not only costly but also reliant on batteries, limiting their applicability [12,13,14]. Therefore, there is an urgent need to find a suitable energy-harvesting method to overcome the drawbacks of conventional power sources and provide a continuous and stable power supply to the sensors. Triboelectric nanogenerators (TENGs) have emerged as an innovative method of energy harvesting, offering affordability and simplicity in fabrication while harnessing mechanical energy from daily environments [15,16,17]. TENGs can also be used in sensors, providing stable voltage signals for applications such as tactile sensing and motion monitoring. Leveraging electrostatic induction, TENG enables non-contact information [18,19]. PDMS has been widely used as the triboelectric layer of TENG in recent years due to its exceptional electrification performance, excellent elasticity, and biocompatibility [20]. However, pure PDMS-based TENGs exhibit suboptimal output performance [21,22], prompting widespread discussion regarding enhancing TENG efficacy. Currently, there are two main methods to improve the performance of TENG, one of which is to fabricate microstructures on the triboelectric material to augment the effective contact area and subsequently enhance the output performance of TENG [23]. For example, Bui et al. created patterned dielectric friction surfaces using highly ordered and non-compactly stacked arrays of microbeads to mimic the friction pads of tree frogs. They achieved a 7-fold increase in power density compared to a TENG based on flat PDMS by tuning the pattern characteristics through adjusting the methanol content [24]. The other method entails mixing the triboelectric material with dielectric or conductive materials, such as Ti_3_C_2_T_x_ and graphene nanosheets [25,26]. Mixing can increase the dielectric constant, resulting in a higher charge density and improved the output performance of TENG [27]. As an example, Gao, Y.Y. et al. developed a MXene-reinforced electret polytetrafluoroethylene (PTFE) film with high mechanical properties and surface charge density by a spraying and annealing treatment. The TENG made from this composite film can deliver an open-circuit voltage of 397 V, a short-circuit current of 21 mA, and a transferred charge of 232 nC, which are 4, 6, and 6 times higher, respectively, than those made from pure PTFE film [28]. Mo_2_CT_x_, a two-dimensional transition metal carbide MXene, is considered as an effective conductive filler due to its excellent electrical conductivity and high specific capacity [29,30]. By mixing Mo_2_CT_x_ into PDMS, the output performance of PDMS can be theoretically improved.

This study presents the development of triboelectric sensors (Mo-TES) using Mo_2_CT_x_/PDMS composite films with varying mass fractions. The output performance of Mo-TES was measured and discussed, as well as its detailed sensing performance as a pressure sensor. In addition, Mo-TES has a non-contact sensing capability, which makes it a promising technology for disease prevention and safety protection.

## 2. Materials and Methods

### 2.1. Preparation of Mo_2_CT_x_

Mo_2_CT_x_ was synthesized through a selective etching process. Firstly, 2 g of LiF was added to 50 mL of concentrated HCl and stirred for 10 min. Subsequently, 1 g of precursor Mo_2_Ga_2_C was gradually dissolved in the solution, and the reaction proceeded until no bubbles were observed in the reactor. Following this, the rotor was removed, and the reaction was etched at 180 °C for 24 h. The reactants obtained from etching were then centrifuged at 6000 rpm for 6 min and washed with deionized water. This centrifugal washing procedure was repeated several times until the supernatant was diluted to neutral.

### 2.2. Preparation of Mo_2_CT_x_/PDMS Composite Films

The preparation process of Mo_2_CT_x_/PDMS composite films is shown in Figure 1a. The Mo_2_CT_x_ solution was dried, grinded into particles and mixed into the PDMS matrix at varying mass fractions (0, 1, 2, 3, 4, and 5 wt%). The curing agent was added at a mass ratio of 1:10 for each sample. The samples were stirred for 30 min, deposited into a special mold (the mold dimension is provided by the Appendix A) and placed in a vacuum oven. Subsequently, air bubbles in the samples were eliminated under a pressure of 80 kPa. The samples were then cured at 80 °C for 4 h to obtain Mo_2_CT_x_/PDMS composite films (20 mm × 20 mm × 0.5 mm).

### 2.3. Fabrication of Triboelectric Sensors Utilizing Mo_2_CT_x_/PDMS Composite Films

As shown in Figure 1b, a layer of conductive copper tape was applied to the backside of the Mo_2_CT_x_/PDMS composite film, followed by the application of a layer of polyethylene terephthalate (PET) tape glued on the copper tape to provide support and protection. The resulting Mo-TES were finally named Mo-TES0, Mo-TES1, Mo-TES2, Mo-TES3, Mo-TES4, and Mo-TES5 based on the mass fraction of mixing Mo_2_CT_x_ particles of 0, 1, 2, 3, 4, and 5 wt%, respectively.

### 2.4. Characterization and Measurement

The crystalline structures were characterized by wide-angle X-ray Diffraction (XRD, Rigaku D/Max 2550) in the 2θ range from 5° to 50° with Cu Kα radiation (λ = 1.5418 Å). The surface morphology and contained elements of Mo_2_CT_x_/PDMS composite films were observed by Scanning Electron Microscope (SEM, Hitachi TM4000Plus and FESEM, JEOL JSM-7900F) with an accelerating voltage of 5 kV. Alpha-High Performance Frequency Analyzer (Alpha-A) was used to test the dielectric constant and dielectric loss of Mo_2_CT_x_/PDMS composite films. Using an auto-injector, 3 μL of deionized water was randomly dropped at different locations on the surface of PDMS and Mo_2_CT_x_/PDMS composite films to investigate the change of water contact angle before and after mixing. The open-circuit voltage and short-circuit current of the Mo-TES were measured using a programmable electrostatic meter (Keithley, 6514), and real-time data recording was facilitated through Keithley KickStart software(KickStartFL-DMM 2.9.0). The electrodynamic measuring table (ESM303, Mark10) was used with the force set to 25 N and the height set to 5 mm in the measurement of Mo-TES with different mixing mass fractions, and the force was set to 100 N with the same height in the measurement of Mo-TES sensing properties. Finite element simulations were implemented by COMSOL (COMSOL 6.0) multi-physics field simulation software.

## 3. Results and Discussion

Firstly, XRD analyses were performed on Mo_2_Ga_2_C and Mo_2_CT_x_ particles, as shown in Figure 2a. Compared with Mo_2_Ga_2_C, the etched Mo_2_CT_x_ shows a new characteristic peak (002), with this peak shifted to a lower angle, implying a reduction in layer spacing and suggesting the successful etching of Ga from Mo_2_Ga_2_C. Figure 2b shows the SEM image of Mo_2_CT_x_ particles. It shows the typical microscopic laminar structure of Mo_2_CT_x_ particles. The microscopic layered structure of Mo_2_CT_x_ is illustrated in Figure 2b. Figure 2c–f shows the surface morphology and structure of pure PDMS films and Mo_2_CT_x_/PDMS composite films with a mixing mass fraction of 3 wt%, respectively. The comparison of the SEM images reveals that the Mo_2_CT_x_/PDMS composite films appear as microscopic laminar structural particles. The energy dispersive spectroscopy (EDS) images of Mo_2_CT_x_/PDMS composite films with different mixing mass fractions are shown in Figure 2d,f and Appendix A. It can be found that the Mo_2_CT_x_/PDMS composite films contain Mo elements that are not originally present, showing that Mo_2_CT_x_ has been mixed in the PDMS matrix.

The working principle of Mo-TES mainly involves contact charging [31] and electrostatic induction [32]. The electrostatic field module of the COMSOL software simulates the Mo-TES process. Figure 3a illustrates the skin, Mo_2_CT_x_/PDMS composite film, and copper electrode, represented by the upper, middle, and lower rectangles, respectively. In the initial state, when the Mo_2_CT_x_/PDMS composite film and the skin are in contact, based on the friction electric series, it is determined that the Mo_2_CT_x_/PDMS composite film is more likely to acquire electrons relative to the skin. As a result, the upper skin surface gathers positive charges while the Mo_2_CT_x_/PDMS composite film surface accumulates an equal amount of negative charges. In this process, the opposite charges on the surfaces of the two films almost coincide in the same plane, so that no potential difference is generated at the copper electrode on the lower part of the Mo_2_CT_x_/PDMS composite film. As shown in Figure 3b, as the skin separates from the Mo_2_CT_x_/PDMS composite film gradually, the potential difference at the copper electrode increases with distance, resulting in free electrons flowing to the ground, there by generating a voltage/current signal output. As shown in Figure 3d, the frontal potential difference on the copper electrodes reached its maximized when the maximum distance is reached. As shown in Figure 3c, as the skin and Mo_2_CT_x_/PDMS composite film approached each other gradually. The potential difference on the copper electrode decreases with decreasing distance, causing electrons to flow from the ground to the copper electrode, thus producing output voltage/current signals with opposite directions. When the skin contacts the Mo_2_CT_x_/PDMS composite film again, the potential difference on the copper electrode decreases to a minimum. Equation (1) provides the open-circuit voltage of TENG, as previously reported [33].
V_oc_ = σ_0_d/ε_0_,(1)

The open-circuit voltage of TENG is denoted by V_oc_, while σ_0_ represents the charge density on the surface of the film. The distance (d) between the surfaces of the two triboelectric materials is represented in the formula, where ε_0_ represents the vacuum dielectric constant. It is evident that the output voltage of Mo-TES increases as the distance between the two triboelectric layers increases for a certain ε_0_ and surface charge. Thus, we investigated the change in potential difference on the copper electrode using the electrostatic field module simulation in COMSOL software. Figure 3e displays the results of the COMSOL simulation. The potential difference on the copper electrode increases as the distance between the epidermis and the Mo_2_CT_x_/PDMS composite film increases, resulting in varying output voltages from the skin at different distances from the Mo-TES.

As shown in Figure 4a,b, the open-circuit voltage and short-circuit current of the Mo-TES0 were approximately 11.7 V and 41.1 nA, respectively. Meanwhile, comparing the Mo-TES mixed with different mass fractions of Mo_2_CT_x_ particles, it was observed that, with the increase in mixing mass fractions, the output performance of the Mo-TES initially increased and then decreases. Specifically, Mo-TES3 exhibited the best performance with a peak open-circuit voltage and short-circuit current of 42.86 V and 201.7 nA, respectively. The peak open-circuit voltage and short-circuit current of Mo-TES3 are 3.66 and 4.91 times higher than those of Mo-TES0, respectively, demonstrating that Mo_2_CT_x_ particles greatly enhance the performance of Mo-TES. It has been previously illustrated that mixing dielectric conductors alters the dielectric constant and dielectric loss of the PDMS films, subsequently affecting the output characteristics of TENG [34,35]. To verify this, we evaluated the dielectric constant and dielectric loss of Mo_2_CT_x_/PDMS composite films by sandwiching the prepared sample films between the test electrodes and measuring the response capacitance at different frequencies using alternating current (AC) electrical ranging from 1 kHz to 1 MHz. As shown in Figure 4c, the dielectric constant of Mo_2_CT_x_/PDMS composite films increases with the rise in the mass fraction of mixing Mo_2_CT_x_ particles across the entire frequency range, indicating that the incorporation of Mo_2_CT_x_ particles enhances the dielectric constant of PDMS films, thereby facilitating the generation of higher surface charge density in the Mo_2_CT_x_/PDMS composite films and consequently enhancing the output characteristics. Furthermore, as shown in Figure 4d, the dielectric loss of the Mo_2_CT_x_/PDMS composite films increases with increasing mass fraction of mixing Mo_2_CT_x_ particles in the whole frequency range. It can be attributed to the aggregation of Mo_2_CT_x_ particles, forming a conductive network in PDMS, which leads to the formation of conductive channels in the Mo_2_CT_x_/PDMS composite films and leads to energy loss in the capacitor device during the external operation, thereby causing in the Mo-TES output performance degradation. It is clearly seen that the combined effect of dielectric constant and dielectric loss contributes to the observed increase followed by a decrease in the output performance of Mo-TES.

The energy harvesting performances of Mo-TES3 under different force conditions (1, 5, 10, 25, 50, and 100 N) were investigated and measured. As shown in Figure 5a,b, both the open-circuit voltage and short-circuit current of Mo-TES3 exhibited an increasing trend with the increase in applied force. Notably, at 100 N, the open-circuit voltage and short-circuit current of Mo-TES3 reached 86.89 V and 578.12 nA, respectively. Furthermore, through the external connection of different matching resistors, we obtained the output voltage and current variations of Mo-TES3, as depicted in Figure 5c. Meanwhile, in Figure 5d, when the external matching resistance is about 200 MΩ, the output power density of Mo-TES3 peaked, reaching a maximum value of 12.45 μW/cm^2^. In addition, by integrating the Mo-TES3 with a full-wave bridge rectifier and a capacitor, the generated AC power can be rectified to direct current (DC) by the rectifier bridge for charging the capacitor. The equivalent circuit is shown in Figure 5e. Subsequently, Figure 5f shows the real-time voltage variation across each capacitor as the Mo-TES3 charges capacitors with different capacitance values (1, 2.2, 4.7, and 10 μF) during a period of 60 s. The continued increase in voltage across the capacitors during the steady tapping of the Mo-TES3 indicates its capabilities to convert the mechanical energy into electrical energy. Table 1 presents a comparison of Mo-TES with other works. In the single-electrode mode of operation, the Mo-TES exhibited superior performance.

The Mo-TES3 serves not only as a power source but also as a sensor for various sensing applications [36,37]. Based on the previous conclusions, it was found that the output voltage of Mo-TES is more stable than its output current. Therefore, Mo-TES selects the output voltage as its sensing signal. It is well known that sensitivity, response recovery time, and stability are important criteria for assessing the sensor performance. Figure 6a shows the relationship between the pressure and output voltage of Mo-TES3. The output voltage of Mo-TES3 slowly increases when the pressure is below 65 kPa and significantly increases when it is above 65 kPa. The sensitivity curve of Mo-TES3 can be divided into two linear regions: 0.041 kPa^−1^ (6.5–65 kPa) and 0.073 kPa^−1^ (65–650 kPa). The coefficients of determination R^2^ for the two linear regions are 0.9726 (6.5–65 kPa) and 0.9745 (65–650 kPa), respectively. These values indicate a good fit and suggest that the output voltage of Mo-TES3 is largely affected by the pressure. In addition, the Appendix A provided the output voltage of Mo-TES0 in relation to pressure, with sensitivities of 0.030 and 0.031 kPa^−1^ in pressure ranges of 6.5–65 and 65–650 kPa, respectively. The comparison results indicate that Mo-TES3 has better sensitivity than Mo-TES0. Next, the response recovery time of Mo-TES3 was tested as shown in Figure 6b. Both the response time and recovery time of Mo-TES3 were around 78 ms, demonstrating its good responsiveness. As shown in Figure 6c, the water contact angle of the Mo_2_CT_x_/PDMS composite film with a mixed mass fraction of 3 wt% was measured to be 109.3° (Appendix A shows the water contact angle for PDMS films and other mixing mass fractions of Mo_2_CT_x_/PDMS composite films), indicating that it has good hydrophobicity. Based on the good hydrophobicity of the Mo_2_CT_x_/PDMS composite film, by comparing the output voltage of Mo-TES3 before and after water washing, it can be found that the output voltage before and after water washing remains around 86 V, which demonstrates its stability in a humid environment. As shown in Figure 6d, the output voltage of Mo-TES3 is basically maintained at around 86 V during 5000 cycles, indicating that Mo-TES3 has fatigue resistance and good stability. The principle of the triboelectric sensor is based on electrostatic induction. According to a previous COMSOL simulation study, the skin can generate output signals at different positions from Mo-TES. Figure 6e shows that a signal can be generated even when the finger has not yet touched Mo-TES3, making it possible to use Mo-TES3 as a non-contact on/off button sensor. As depicted in Figure 6f, Mo-TES3 is mounted on an elevator button to enable users to activate it without physical contact, thereby reducing the risk of disease transmission. Additionally, Mo-TES3 can assist users who have injured their fingers and have difficulty operating the buttons.

**Table 1 nanomaterials-14-00428-t001:** Performance comparison between this work and others.

Reference	Mode	Material	Size	Stresses	Frequency	Voltage	Power Density
[38]	Single electrode	Hydrogel	20 mm × 20 mm	0.67 kPa	1 Hz	≈71 V	-
[39]	Single electrode	CuNWs/PDMS	20 mm × 20 mm	-	2 Hz	≈45 V	13.4 μW/cm^2^
[40]	Single electrode	Carbon fiber/PDMS	-	50 N	1 Hz	≈72 V	74.1 μW/cm^2^
[41]	Single electrode	BaTiO_3_/PDMS	30 mm × 30 mm	-	1 Hz	72.2 V	-
[42]	Single electrode	MXene/PDMS	-	30 kPa	3 Hz	≈70 V	-
[43]	Single electrode	Graphene oxide/PDMS	-	4 N	4 Hz	79.4 V	-
[44]	Single electrode	Copper foam/PDMS	-	15 N	3 Hz	≈15 V	-
[45]	Single electrode	Fluorinated carbon black/PDMS	-	-	-	≈60 V	-
This work	Single electrode	Mo_2_CT_x_/PDMS	20 mm × 20 mm	100 N	1 Hz	86.89 V	12.45 μW/cm^2^

## 4. Conclusions

In summary, the triboelectric sensor (Mo-TES) with a Mo_2_CT_x_/PDMS composite film as the triboelectric layer has been developed and fabricated. With the increase in the mixing mass fraction of Mo_2_CT_x_ particles, the output performance of Mo-TES increases but then decreases due to the combined effect of the dielectric constant and dielectric loss. The output performance of Mo-TES3 reaches the optimum when the mixing mass fraction of Mo_2_CT_x_ particles is 3 wt%. Furthermore, the output performance of Mo-TES3 increases proportionally with the applied forces, and the output voltage and current of Mo-TES3 reach 86.89 V and 578.12 nA, respectively, at a force of 100 N. Additionally, the maximum output power density is 12.45 μW/cm^2^ with a matching resistance of 200 MΩ. Moreover, Mo-TES3 demonstrates excellent sensitivity and an extremely fast response and recovery speed. In addition, Mo-TES3 exhibits exceptional stability in both the water washing and 5000 cycles anti-fatigue tests. Mo-TES3 can realize non-contact sensing, and when applied to public facilities, it can reduce people’s interactive contact and spread of diseases.

## Figures and Tables

**Figure 1 nanomaterials-14-00428-f001:**
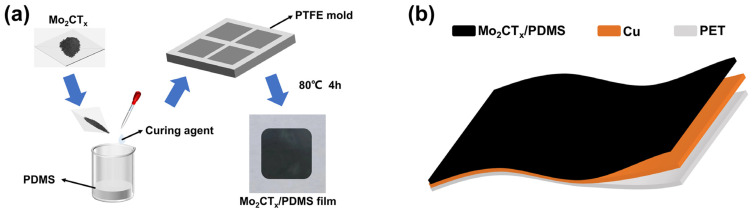
The preparation of the material and the assembly of the Mo-TES: (**a**) Fabrication process of Mo_2_CT_x_/PDMS composite films. (**b**) Assembly of the Mo-TES.

**Figure 2 nanomaterials-14-00428-f002:**
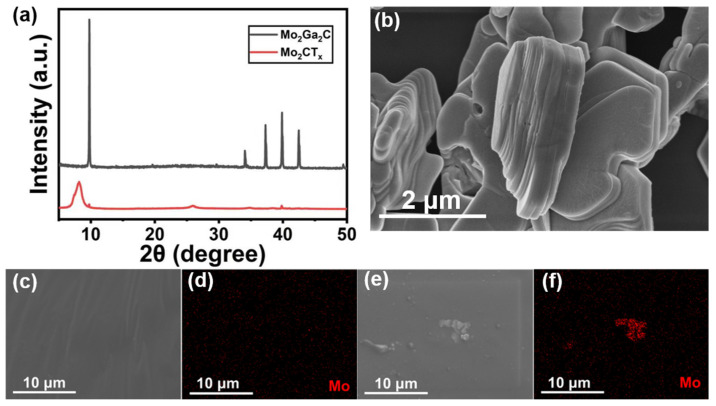
Characterization of Mo_2_CT_x_ and Mo_2_CT_x_/PDMS composite films: (**a**) The XRD image of Mo_2_Ga_2_C and Mo_2_CT_x_. (**b**) The SEM image of Mo_2_CT_x_. (**c**,**d**) The EDS image of pure PDMS film. (**e**,**f**) The EDS images of Mo_2_CT_x_/PDMS composite films mixed with Mo_2_CT_x_ particles with a mass fraction of 3 wt%.

**Figure 3 nanomaterials-14-00428-f003:**
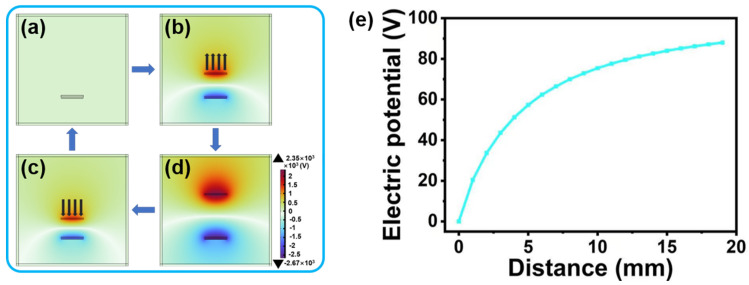
Working principle of the Mo-TES: (**a**–**d**) COMSOL simulation of the working principle of the Mo-TES. (**e**) COMSOL simulation of the potential difference change on the copper electrode.

**Figure 4 nanomaterials-14-00428-f004:**
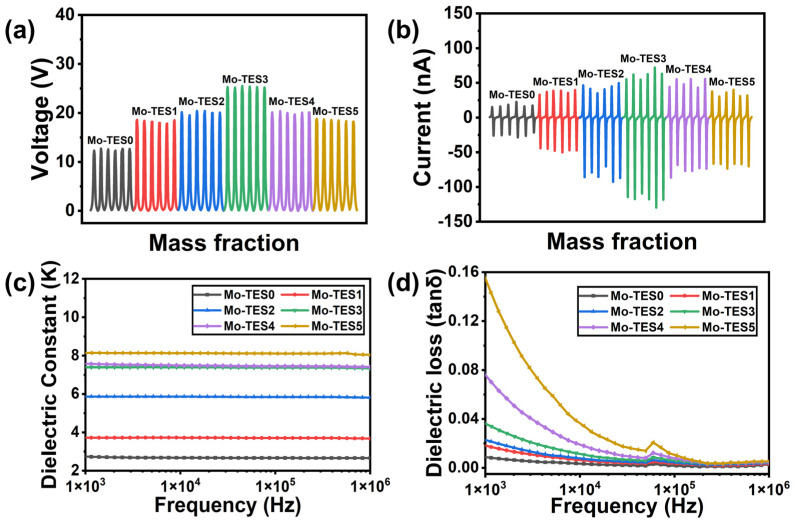
The output characteristics of the Mo-TES mixed with different mass fractions of Mo_2_CT_x_ particles: (**a**) The open-circuit voltage of the Mo-TES mixed with different mass fractions of Mo_2_CT_x_ particles. (**b**) The short-circuit current of the Mo-TES mixed with different mass fractions of Mo_2_CT_x_ particles. (**c**) The dielectric constant of the triboelectric layer of the Mo-TES mixed with different mass fractions of Mo_2_CT_x_ particles. (**d**) The dielectric loss of the triboelectric layer of the Mo-TES mixed with different mass fractions of Mo_2_CT_x_ particles.

**Figure 5 nanomaterials-14-00428-f005:**
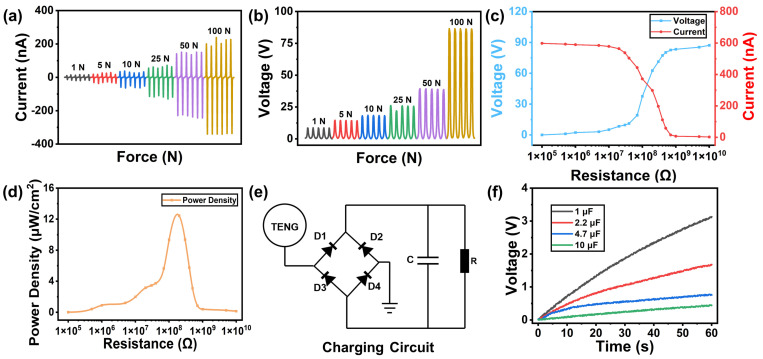
The output performance of the Mo-TES3: (**a**) The open-circuit voltage of the Mo-TES3 with different forces. (**b**) The short-circuit current of the Mo-TES3 with different forces. (**c**) The output current and voltage of the Mo-TES3 with different external matching resistors. (**d**) The output power density of the Mo-TES3 with different external matching resistors. (**e**) The charging circuit of the Mo-TES3. (**f**) Within 60 s, the Mo-TES3 charges capacitors of different capacitance values.

**Figure 6 nanomaterials-14-00428-f006:**
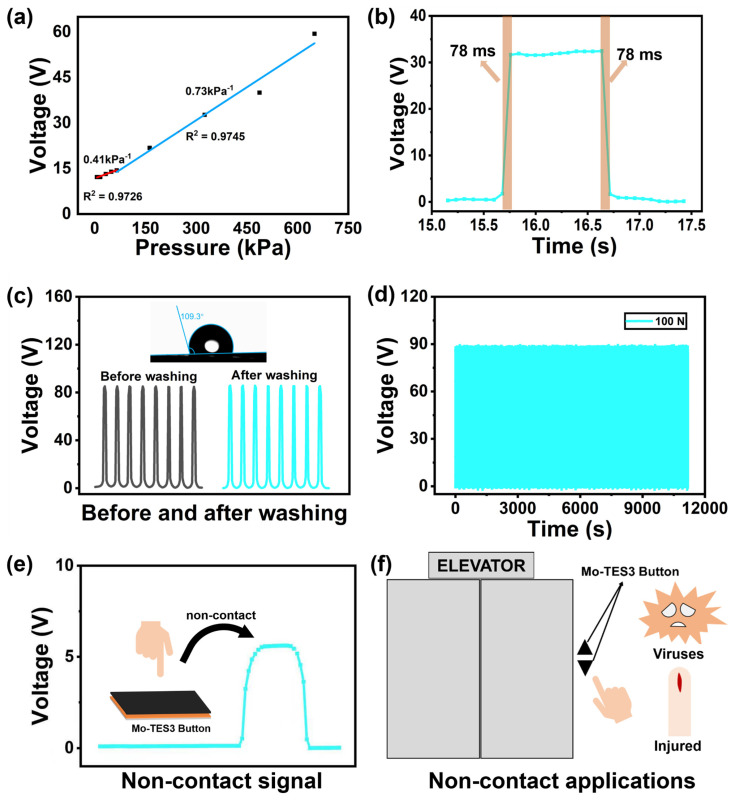
The sensing properties of the Mo-TES3 and its applications: (**a**) The output voltage and pressure relationship curve of the Mo-TES3. (**b**) The response recovery time of the Mo-TES3. (**c**) The output voltage of the Mo-TES3 before and after water washing. The water contact angle images of Mo_2_CT_x_/PDMS composite films mixed with Mo_2_CT_x_ particles with a mass fraction of 3 wt%. (**d**) The output voltage of the Mo-TES3 within 5000 cycles. (**e**) The output voltage of the Mo-TES3 button in non-contact application. (**f**) Non-contact application of the Mo-TES3.

## Data Availability

Data are contained within the article.

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
