# Peer review of "Highly Sensitive and Stable Multifunctional Self-Powered Triboelectric Sensor Utilizing Mo2CTx/PDMS Composite Film for Pressure Sensing and Non-Contact Sensing"

_nanomaterials, 2024, doi:10.3390/nano14050428_

Round 1

Reviewer 1 Report

Comments and Suggestions for Authors

Comments to the authors-Manuscript ID: nanomaterials-2873009

In the present work entitled “Highly Sensitive and Stable Triboelectric Sensor Utilizing Mo2CTx/PDMS Film for Non-Contact Sensing and Recognition Applications”, the authors have made a thorough study regarding the topic. The data acquisition is quite elaborate and informative for the readers. However, there are few points that need to be clarified for acceptance of the article: -

1.     Line 188-190, “By calculating the obtained sensitivity Sn (ratio of output range voltage to input range pressure), we obtain sensitivity values of 0.032, 0.052, 0.073, 0.111, 0.081 and 0.069 for Mo-TES0 to Mo-TES5, respectively.” To increase the readability and understanding the actual sensitivity without any ambiguity, sensitivity should be better represented in the standard way, both in graphic and text.

2.     Though the authors have claimed its application in non-contact sensing (e.g. elevator switch as per Fig. 5e), but how much is the probability of real-field application with such low power harvesting? The output voltage may be high but the current is too low to create a sufficient power density.

3.     In Figure 5f, it is shown that the Mo-TES3 generates an output signal even before direct contact with the skin, this information is quite misleading! What is the distance of the finger from the device? Should be given a real-time image to show.

Author Response

Thank you for your constructive suggestions! Please see the attachment.

Reviewer 2 Report

Comments and Suggestions for Authors

In this work, the authors present a sensor with high sensitivity and fast response based on Mo2CTx/PDMS triboelectric nanogenerators (TENGS). The sensor is stable in water washing and antifatigue testing. It also shows the potential of noncontact sensing for disease prevention and safety protection. The results are novel and interesting. The paper is well organized. The following issues should be clearly addressed before it can be accepted.

1.         On line 94 “Firstly, characterization of Mo2CTx and Mo2CTx/PDMS films was measured and shown in Figure 2a.” However, the Figure 2a is the XRD of Mo2Ga2C and Mo2CTx. The description is not matched.

2.         The author should give SEM and TEM images of Mo2CTx to show its morphology.

3.         The author give Figure 2j to show the simulation of the potential difference change. How did the distance affect the true output voltage and current?

4.         Did the Mo2CTx addition affect the surface potential? The Kelvin Probe testing can help to measure the surface potential with different percentages of Mo2CTx.

5.         On line 151, the author said “forming a conductive network in PDMS”, what is the resistance or conductivity for the PDMS film and Mo2CTx/PDMS films?

6.         How big of the force applied by the finger in Figure 5f?

7.         The response time and recovery time of Mo-TES3 were around 78 ms. Will the time be affected by the time resolution? The author is better to decrease the time interval to get a precise response time and recovery time.

8.         Will the non-contact distance affect the signal? Please show evidence.

9.         Will the area of the finger area affect the result? Will the adult or child finger get the same output voltage?

10.       As a sensor, it must have a calibration curve with pressure sensitivity. Figure 5a should be modified.

11.       Only 1000 cycles for stability were reported. It is too short for the stability test. How about 10000 cycles?

12.       Figure 5e is not suitable for showing in the manuscript. It is better in supporting material or TOC.

Comments on the Quality of English Language

None

Author Response

(The authors gave the same response as above.)

Reviewer 3 Report

Comments and Suggestions for Authors

This study presents the development of a Mo2CTx-based triboelectric sensor (Mo-TES), utilizing Mo2CTx/polydimethylsiloxane (PDMS) film, with a focus on optimizing the mass fraction and force of Mo2CTx particles to enhance sensor performance. The name suggests a "non-contact sensing", however the whole manuscript does not sufficiently give details what it is and how it works, introduction is missing the explanation of work mechanism, conclusions are just naming the non-contact applications.

The authors mention just the ultrasonic and laser detection sensors as the non-contact examples, however the most used alternative, the capacitive sensors are not even mentioned. I suggest to the authors to consider the most used sensors for proximity, such as inductive, capacitive, photoelectric.

Moreover it is important to consider the mechanism of reported triboelectric non-contact sensor in light of other mentioned proximity sensor mechanisms, and change the explanation accordingly.

Comments on the Quality of English Language

Already the abstract has grammar issues, i.e. "When Mo2CTx particles is 3 wt%, Mo-TES3 achieves an open-cir-9cuit voltage up to 86 V, a short-circuit current of up to 578 nA, a power density of up to 12.45 10μW/cm2" has 2 mistakes, verb missing.

Author Response

(The authors gave the same response as above.)

Round 2

Reviewer 2 Report

Comments and Suggestions for Authors

None

Comments on the Quality of English Language

None